# Prenatal Predictors and Early Postnatal Outcomes in Fetuses Diagnosed with Tricuspid Atresia

**DOI:** 10.3390/diagnostics14242855

**Published:** 2024-12-19

**Authors:** Ozge Kahramanoglu, Omer Gokhan Eyisoy, Oya Demirci

**Affiliations:** Zeynep Kamil Women and Children’s Diseases Training and Research Hospital, Department of Perinatology, Health Science University, Istanbul 34668, Turkey; dr.gokhaneyisoy@gmail.com (O.G.E.); demircioya@gmail.com (O.D.)

**Keywords:** Doppler, echocardiography, pulmonary stenosis, tricuspid atresia, ventricular septal defect

## Abstract

**Objective:** To assess the prenatal course and early postnatal outcomes of fetuses diagnosed with tricuspid atresia and to identify predictors of survival. **Methods:** This was a retrospective study of 25 fetuses diagnosed with tricuspid atresia in a single tertiary referral center, evaluating prenatal echocardiographic features and postnatal outcomes. **Results:** A total of 4 of 29 initially diagnosed fetuses were excluded due to changes in diagnosis or loss to follow-up, leaving 25 fetuses for analysis. Of these, 16 (64%) had concordant VA alignment, 8 (32%) had discordant VA connections, and 1 had a double-outlet left ventricle (DOLV). Pulmonary stenosis was observed in 13 fetuses, and 10 (40%) had extracardiac anomalies. Genetic testing, performed in 5 cases, identified a chromosomal anomaly in one case (trisomy 18). Overall, three pregnancies were terminated due to severe associated anomalies. Among the 22 liveborn infants, survival at 12 months was 72%. Restrictive ventricular septal defect (VSD) and the high ductus venosus pulsatility index were significantly associated with lower survival (*p* = 0.021 and *p* = 0.034, respectively). **Conclusions:** Tricuspid atresia can be accurately diagnosed in utero with a thorough echocardiographic evaluation. Restrictive VSD and outflow tract obstructions are critical determinants of early survival, while abnormal DV Doppler patterns may serve as additional markers for adverse outcomes. More extensive studies are needed to validate these findings and improve prognostic counseling.

## 1. Introduction

Fetal tricuspid atresia (TA) is a rare and complex congenital heart defect characterized by the complete absence of the tricuspid valve, which results in an obstruction of blood flow from the right atrium to the right ventricle, leading to significant hemodynamic alterations [1,2]. This anomaly accounts for approximately 3% of all congenital heart defects and poses considerable challenges to prenatal diagnosis and management due to its varied anatomical presentations and associated anomalies [3,4]. These associated cardiac anomalies can significantly impact prognosis, making the assessment of the right ventricular function and associated defects a crucial aspect of prenatal evaluation.

The absence of the tricuspid valve typically results in an underdeveloped right ventricle, often accompanied by a ventricular septal defect (VSD), and is frequently associated with additional anomalies such as pulmonary stenosis or atresia [5]. Pulmonary outflow obstruction, found in up to 75% of cases, is the most prevalent associated anomaly [6]. This can arise from small VSDs, pulmonary valve stenosis, or right ventricular outflow tract hypoplasia. Other cardiac anomalies, such as ventriculoarterial discordance, transposition of the great arteries, or aortic coarctation, further adversely affect the hemodynamics of the fetal circulation, which in turn may compromise postnatal adaptation and necessitate early surgical intervention [7,8].

In addition to cardiac anomalies, extracardiac malformations are also commonly associated with TA and may further complicate prognosis and management [9]. These extracardiac anomalies can include chromosomal abnormalities, central nervous system malformations, and other structural defects, all of which can influence the prenatal course and postnatal outcomes [10]. Therefore, comprehensive evaluation of extracardiac structures is essential in the prenatal assessment of fetuses with TA, as it may guide decision making regarding pregnancy continuation, delivery planning, and postnatal intervention. Despite advances in fetal echocardiography, the available data on the spectrum and long-term outcomes of TA remain scarce. While prior studies have focused on isolated aspects such as prenatal diagnosis or specific postnatal outcomes, comprehensive evaluations that address the full spectrum of prenatal findings and their impact on early postnatal outcomes are limited [11,12]. This study aims to fill these gaps by analyzing the prenatal characteristics and short-term consequences of tricuspid atresia cases in a single center, providing insights that could guide both prenatal counseling and postnatal outcomes.

## 2. Material and Methods

### 2.1. Study Design and Patient Selection

This retrospective single-center study analyzed the medical records of fetuses diagnosed with tricuspid atresia at our institution between January 2014 and December 2022. Ethical approval for the study was obtained from the Ethics Committee of the University of Health Science, Zeynep Kamil Women and Children’s Diseases Training and Research Hospital (Approval number: [23.11.2022-130]). All procedures adhered to the tenets of the Declaration of Helsinki, and informed consent was obtained for all examinations.

All fetuses diagnosed with tricuspid atresia during routine second-trimester screening or referred to our center for a suspected cardiac anomaly were included in the study. This approach ensured that both newly identified cases and those previously evaluated elsewhere were comprehensively assessed. The inclusion process relied on detailed fetal echocardiography performed by experienced fetal cardiologists to confirm the diagnosis of TA and assess associated anomalies. Exclusion criteria consisted of cases with missing or incomplete clinical records, which could compromise the accuracy of data analysis, as well as cases lost to follow-up, where outcome data could not be verified. Additionally, fetuses with postnatal diagnoses that were not confirmed by a pediatric cardiologist were excluded to maintain the reliability and consistency of our cohort, ensuring that only confirmed cases of tricuspid atresia were analyzed.

### 2.2. Prenatal Assessment and Echocardiographic Evaluation

Fetal echocardiographic evaluations were performed by the same team of experienced perinatologists using a Voluson E6 ultrasound system (GE Medical Systems, Milwaukee, WI, USA) with a 2–8 MHz transducer. Each examination involved a detailed assessment of the fetal cardiac structure, focusing on critical aspects such as the characterization of isolated tricuspid atresia, identification of restrictive ventricular septal defects (VSDs), and evaluation of atrioventricular (AV) and ventriculoarterial (VA) alignments. The restrictive nature of the VSD was defined based on its influence on intracardiac hemodynamics, including right ventricular performance and shunt flow patterns across the defect. Doppler parameters such as the Ductus Venosus Pulsatility Index (DV-PI), Umbilical Artery PI (UA-PI), and Middle Cerebral Artery PI (MCA-PI) were systematically recorded during each follow-up visit to monitor hemodynamic changes and placental resistance.

During follow-up, the frequency of ultrasonographic assessments varied depending on gestational age. Monthly evaluations were conducted until 36 weeks, after which the frequency increased to weekly visits until delivery. More frequent evaluations were performed in cases of suspected hemodynamic instability or progression of cardiac anomalies to assess fetal well-being and the need for possible early intervention.

### 2.3. Data Collection and Variable Definitions

Electronic medical records were reviewed to extract detailed maternal demographics, including age and relevant health history, gestational age at diagnosis, and comprehensive echocardiographic findings. All associated cardiac and extracardiac anomalies were documented to provide a thorough overview of each case. Postnatal data, including gestational age at delivery, birth weight, and early survival outcomes, were also systematically recorded to assess both immediate and longer-term postnatal outcomes.

### 2.4. Statistical Analysis

Continuous variables were reported as mean ± standard deviation (SD) or median with interquartile range (IQR) based on the data distribution. Categorical variables were expressed as frequencies and percentages. Comparative analyses between groups were conducted using independent *t*-tests or Mann–Whitney U tests for continuous variables and Chi-square or Fisher’s exact tests for categorical variables, as appropriate. A *p*-value < 0.05 was considered statistically significant. All statistical analyses were performed using SPSS version 25.0 (IBM Corp., Armonk, NY, USA).

## 3. Results

A total of 29 fetuses were initially identified with a prenatal diagnosis of tricuspid atresia (TA) between 12 and 40 weeks of gestation. After excluding three fetuses with a subsequent change in diagnosis (pulmonary atresia and intact ventricular septum in two and double-inlet left ventricle in one) and one case that was lost to follow-up, the final cohort comprised 25 fetuses, with a median gestational age at diagnosis of 26 weeks (range, 12–40 weeks) (Figure 1).

The median maternal age was 32 (range, 21–37) years. Of the 25 fetuses, 16 (64%) were male and 9 (36%) were female. Fetal growth restriction (FGR) was identified in seven cases (28%), though all affected fetuses were delivered beyond 36 weeks. Genetic testing was performed in five cases, revealing a chromosomal anomaly in one case (trisomy 18), while the remaining four tests yielded normal results. The median gestational age at delivery was 36.5 weeks (range, 28–40 weeks), and the median birth weight was 2480 g (range, 1110–3360 g) (Table 1).

Segmental analysis revealed that the majority of fetuses (*n* = 24) had atrial situs solitus, while only one fetus exhibited dextrocardia. Atrioventricular (AV) and ventriculoarterial (VA) concordance was noted in 16 cases, whereas eight fetuses demonstrated AV concordance with VA discordance. A single case was identified as having a double-outlet left ventricle (DOLV). Outflow tract abnormalities were common, with pulmonary stenosis observed in 13 fetuses, aortic hypoplasia in two cases, and coarctation of the aorta in one case. Additional cardiac anomalies included the presence of a left superior vena cava (LSVC) in two cases and a mitral valve cleft with regurgitation in one fetus (Table 2).

Fifteen fetuses (60%) had isolated TA without any accompanying anomalies, while the remaining ten fetuses (40%) exhibited a range of extracardiac malformations. The most frequently observed anomalies included hyperechogenic bowel in two fetuses and renal pelviectasia in two cases. Additionally, single cases of Dandy–Walker malformation, single umbilical artery, ventriculomegaly, and cystic hygroma were documented. Musculoskeletal anomalies were also noted, with two fetuses presenting with pes equinovarus.

Table 3 compares Doppler and echocardiographic indices between the survivors and non-survivors. Restrictive ventricular septal defect (VSD) and restrictive foramen ovale were significantly more prevalent in the non-survivor group (*p* = 0.021 and *p* = 0.043, respectively). Similarly, pulmonary stenosis was found in a higher proportion of non-survivors than survivors (88.9% vs. 31.3%, *p* = 0.014), suggesting a critical role of outflow tract obstruction in determining outcomes. The ductus venosus pulsatility index (PI) was elevated in non-survivors (1.34 vs. 0.998, *p* = 0.034), while none of the other Doppler parameters differed between groups. All cases diagnosed with tricuspid atresia, even those that later underwent termination of pregnancy, were included in the analysis of Doppler indices at the time of diagnosis (Table 3). This approach was used to ensure a comprehensive representation of the cohort. While the gestational ages at the time of Doppler assessment may not have been available for all TOP cases, their inclusion provides an unbiased depiction of Doppler findings across the full spectrum of diagnosed cases. In our comparison between survivors and non-survivors, we evaluated gestational age at delivery and birth weight to assess their potential impact on early neonatal survival. The analysis showed no statistically significant difference between the two groups, with *p*-values greater than 0.05 for both parameters. These findings suggest that neither gestational age at delivery nor birth weight was a significant predictor of survival within our cohort (Table 4).

The study cohort comprised 25 fetuses, categorized into groups based on ventriculoarterial (VA) alignment: 16 with concordant VA and 8 with discordant VA connections, while 1 case presented with a double-outlet left ventricle (DOLV). Termination of pregnancy (TOP) was performed in three cases due to severe associated anomalies, including Dandy–Walker malformation, trisomy 18, and anhydramnios due to PPROM (premature rupture of membranes). Among the remaining 22 liveborn infants, 6 died during the neonatal period. One infant, delivered at a gestational age of 37 weeks and 2 days with a birth weight of 2970 g, passed away on the fourth postpartum day. Another infant, born at 38 weeks with a birth weight of 2180 g, died on the 20th postpartum day. Additionally, an infant delivered at 36 weeks and 5 days died on the 49th postpartum day, while another, born at 36 weeks and 4 days weighing 2050 g, passed away during the third month of life. An infant delivered at 30 weeks of gestation, weighing 1110 g, died on the 10th postpartum day. Finally, an infant born at 36 weeks with a birth weight of 2400 g passed away on the ninth postpartum day. The survival rates at 3 and 12 months were significantly influenced by the presence of pulmonary stenosis. No intrauterine fetal demise (IUFD) occurred. For fetuses with concordant VA and no associated pulmonary stenosis, survival at 12 months was 100%, whereas only 40% of those with pulmonary stenosis survived beyond the neonatal period. In the discordant VA group, 75% survived at 12 months, with the most favorable outcomes observed in the absence of significant outflow obstruction (Table 5).

## 4. Discussion

A prenatal diagnosis of tricuspid atresia (TA), including its associated intra- and extracardiac anomalies, can be achieved with a high degree of accuracy when employing comprehensive fetal echocardiographic techniques. Consistent with previously published data, the diagnostic accuracy of prenatal TA remains robust. In the series by Wald et al., the postnatal diagnosis diverged from the prenatal findings in only 3 out of 91 cases (3.3%), demonstrating the reliability of detailed prenatal assessment [11]. In our study, all diagnoses were confirmed postnatally by a pediatric cardiologist, and no discrepancies were identified, indicating a high concordance between prenatal and postnatal diagnoses. The high accuracy in diagnosis observed in our cohort may reflect the detailed protocol we followed, including segmental analysis and comprehensive Doppler studies. Despite these advancements, the condition’s rarity continues to limit the available data on long-term outcomes and prognostic factors. Studies such as those by Berg et al. and Sharland et al. have documented the prenatal course and outcomes of TA; however, many reports lack granular details on associated anomalies and long-term follow-up beyond the neonatal period [3,13]. The study by Wald et al., encompassing 88 cases over a 15-year period from three major centers, remains one of the most comprehensive references for parental counseling and outcome prediction in fetuses diagnosed with TA [11]. Consequently, multicenter collaborations are needed to accumulate larger sample sizes, which could enhance the understanding of TA’s full spectrum and associated anomalies.

The ventriculoarterial (VA) discordance rate in our cohort was 32%, notably lower than the incidence reported by Wald et al. and Berg et al. but remaining higher than that described by Sharland [3,11,13]. The differences in the VA discordance rates among various studies could be attributed to the diversity in patient populations and the time frame over which these studies were conducted. VA discordance often introduces complexity to the overall hemodynamics, affecting systemic and pulmonary blood flow. Our finding that all cases with aortic arch hypoplasia or outflow tract obstruction were associated with discordant VA alignments supports the notion that VA discordance may predispose one to aortic arch abnormalities. Compared to other cohorts, such as those reported by Berg et al., the incidence of aortic arch hypoplasia in our study was relatively lower (8%), which may be attributed to our smaller sample size [13]. This discrepancy highlights the importance of sample size in rare congenital anomalies, as even small numbers can significantly alter the observed prevalence of certain anatomical patterns.

Our analysis of genetic abnormalities also revealed some differences from previous studies. In our study, genetic testing was performed on only 5 out of 25 fetuses, identifying one case with trisomy 18, while the remaining four tests were normal. This rate is considerably lower than Berg et al., who reported a higher prevalence of chromosomal abnormalities in their cohort [13]. This is in contrast to Wald et al., where chromosomal anomalies were identified in 3.4% of liveborn infants, and an additional 3.4% had syndromic associations, highlighting a potentially underreported prevalence due to incomplete genetic testing and early pregnancy terminations [11]. Moreover, the relatively low rate of chromosomal anomalies observed in our cohort could be attributed to the limited number of cases that underwent genetic evaluation. The increasing accessibility and affordability of genetic testing in recent years provide a valuable opportunity to enhance the prenatal evaluation of fetuses with tricuspid atresia. However, we acknowledge that the number of cases that underwent genetic testing in our study was limited, which restricts our ability to fully assess the genetic contributions to prognosis. We recommend that future research include routine genetic screening of all cases diagnosed with tricuspid atresia. Routine genetic testing, which is now much more accessible than in the past, could aid in more precise risk stratification and in identifying associated anomalies, thereby improving prenatal counseling and individualized management plans. Given the potentially significant impact of chromosomal anomalies on prognosis, it is crucial to offer comprehensive genetic evaluation, including advanced methods like chromosomal microarray analysis and whole exome sequencing, to all fetuses diagnosed with TA. Future studies should emphasize the importance of genetic evaluation as a part of routine prenatal assessment in congenital heart disease.

In our study, ductus venosus PI was significantly different between the survivor and non-survivor groups, indicating that abnormal venous Doppler patterns may be a marker of poor prognosis in fetuses with tricuspid atresia. This finding contrasts with previous studies by Berg et al. and Gembruch et al., where alterations in DV flow were not correlated with fetal outcome, suggesting that in isolated TA cases, these Doppler changes primarily reflect the unique hemodynamics of the defect rather than fetal compromise [14,15]. In Berg et al.’s cohort, DV flow alterations were attributed to increased right atrial pressure and restricted foramen ovale flow, leading to a compensatory shunting mechanism across the atrial septum without compromising fetal well-being [14]. In our cohort, fetuses with foramen ovale restriction also exhibited abnormal DV flow, which may have influenced our results [16]. The concurrent presence of restricted foramen ovale and abnormal DV flow could indicate increased right atrial pressures, leading to altered fetal hemodynamics [17]. However, the small sample size limits the strength of these observations. Similarly to our study, Bianco et al. demonstrated that abnormal DV Doppler patterns, including increased PI and reversed flow, were associated with decreased survival in a broader spectrum of right heart obstructive lesions, such as pulmonary atresia or Ebstein’s anomaly, where right ventricular function is a critical determinant of outcome [18,19,20]. This underlines the value of incorporating DV flow assessment into the routine evaluation of fetuses with TA, as it may provide crucial insights into fetal hemodynamic compromise and the need for targeted perinatal intervention.

The disparity in findings between our study and others may be explained by the distinctive hemodynamic profile of TA, wherein the severity of right heart obstruction and the ability to maintain adequate systemic output through the left heart play a crucial role in determining perinatal outcomes. In conditions like Ebstein’s anomaly, DV flow reversal is often a direct consequence of elevated right atrial pressure due to severely impaired right ventricular filling and function, which is not typically seen in isolated TA unless accompanied by additional intracardiac or extracardiac anomalies [21,22]. Therefore, the predictive value of DV flow in TA may require careful interpretation alongside other hemodynamic parameters to predict outcomes accurately. This underscores the need for further research to delineate the specific circumstances under which DV Doppler indices can serve as reliable markers of fetal compromise in right heart obstructive lesions.

In our cohort, the overall survival rate was 64%, while the termination rate was 12%. Among liveborn infants, the survival rate at one year was 72%, indicating that early termination significantly impacted overall survival rates. These figures are slightly lower compared to prior reports [23] but were higher than that reported by Allan et al., who found a survival rate of 61% for liveborns [24]. For instance, Lan et al. reported a higher first-year survival rate of 68% in fetuses with outflow tract obstructions, which contrasts with our findings of significantly lower survival rates in patients with restrictive VSD and outflow tract obstructions [25]. In our cohort, these specific anatomical features were associated with markedly poorer outcomes, underscoring their critical impact on prognosis. Restrictive VSDs are known to limit effective blood flow between the ventricles, increasing pressure load on the right atrium and impairing cardiac output. This finding aligns with previous studies that have proposed restrictive VSDs as significant predictors of adverse outcomes in congenital heart disease due to their impact on ventricular hemodynamics [26]. Similarly, outflow tract obstructions have been previously reported as key determinants of survival, particularly in the context of right heart obstructive lesions [27]. The lower survival observed in our series may reflect the hemodynamic burden imposed by restrictive VSDs and the severity of outflow obstruction, compromising effective ventricular filling and systemic circulation. These findings suggest that restrictive intracardiac flow patterns, combined with complex outflow tract abnormalities, are critical determinants of survival, contributing to the discrepancy with other studies. Furthermore, pulmonary outflow obstruction, which was present in over half of our cases, further compromises the efficiency of pulmonary circulation, leading to increased strain on the heart and subsequent risk of heart failure. A more aggressive perinatal management approach, such as early intervention to address restrictive VSDs or pulmonary blood flow, could potentially improve outcomes in these high-risk cases.

While this study provides valuable insights into prenatal predictors and early postnatal outcomes in fetuses diagnosed with tricuspid atresia, the findings underscore the need for further research to address several gaps. Future studies should prioritize the evaluation of long-term outcomes, including survival beyond the neonatal period and developmental milestones, to provide a more holistic understanding of prognosis. Additionally, expanding the scope of genetic testing in such cohorts will be instrumental in identifying underlying chromosomal or syndromic associations, facilitating more personalized prenatal counseling and management strategies. Multicenter collaborations with larger sample sizes and standardized protocols will be essential to validate our findings and advance the field.

The main limitations of this study are the small sample size because of the rarity of the disease and its retrospective nature. Due to the retrospective design, we were unable to collect data on surgical procedures and postoperative complications, which limits our understanding of the full spectrum of factors influencing patient outcomes. In addition, the relatively short follow-up period restricts our ability to assess long-term survival and developmental outcomes, which are crucial for a comprehensive understanding of the prognosis in fetuses with tricuspid atresia.

## 5. Conclusions

Our study provides valuable insights into the prenatal predictors and early postnatal outcomes in fetuses diagnosed with tricuspid atresia. We found that factors such as restrictive ventricular septal defects and outflow tract obstructions were strongly associated with decreased survival, highlighting the critical role of detailed prenatal evaluation in risk stratification. Larger prospective multicenter studies with extended follow-up periods are needed to confirm these results and optimize prenatal counseling and postnatal management for affected families.

## Figures and Tables

**Figure 1 diagnostics-14-02855-f001:**
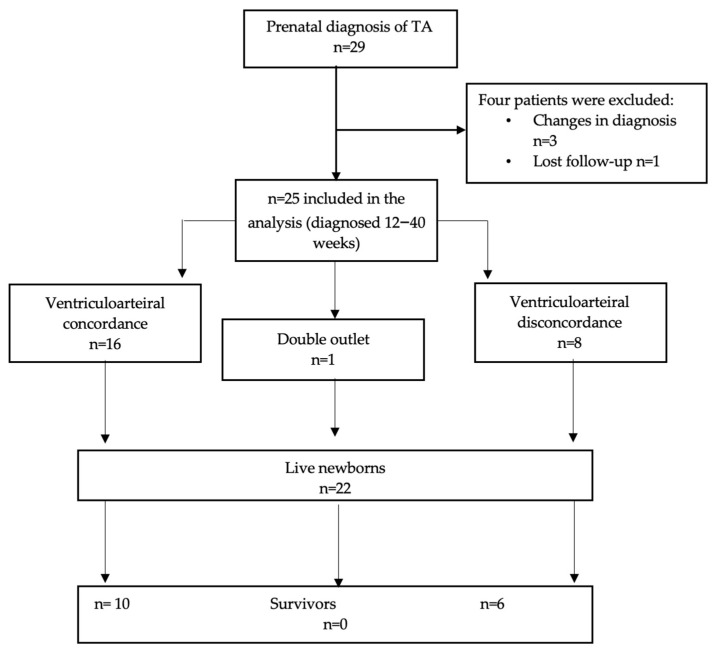
Flowchart of the study group.

**Table 1 diagnostics-14-02855-t001:** Demographic and clinical characteristics of fetuses diagnosed with Tricuspid Atresia.

	All Patients with TA (*n* = 25)
Maternal age, years median	32 (21–37)
Gestational age at diagnosis, weeks median	26 (12–40)
Gestational age at delivery, (weeks) median	36.5 (28–40)
Median birth weight, gr	2480 (1110–3360)
Gender	
- Male, *n* (%)	16 (64)
- Female, *n* (%)	9 (36)
Fetal Growth Restriction (FGR), *n* (%)	7 (28%)
Isolated TA, *n* (%)	15 (60%)
Extracardiac anomalies, *n* (%)	10 (40%)
Genetic Findings	
- Performed	5
- Normal	4
- Chromosomal abnormality	1
- Not performed	20

TA: Tricuspid atresia.

**Table 2 diagnostics-14-02855-t002:** Cardiac structure of patients with TA.

Segmental Analysis		
	Atrial situs solitus, *n* (%)	24 (96%)
	Dextrocardia, *n* (%)	1 (4%)
	AV concordance VA concordance, *n* (%)	16 (64%)
	AV concordance VA discordance, *n* (%)	8 (32%)
	Double outlet (DOLV), *n* (%)	1 (4%)
Outflow obstruction		
	Pulmonary stenosis	13 (52%)
	Aortic hypoplasia	2 (8%)
	Coarctation of aorta	1 (4%)
Additional cardiac abnormalities		
	LSVC	2 (8%)
	Mitral valve cleft with insufficiency	1 (4%)

AV: Atrioventricular, VA: Ventriculoarterial, DOLV: Double-outlet left ventricle, LSCV: Left superior vena cava.

**Table 3 diagnostics-14-02855-t003:** Doppler indices at the time Tricuspid Atresia was diagnosed.

	Survivors (*n* = 16)	Non-Survivors (*n* = 9)	*p* Value
Isolated TA, *n* (%)	11 (68.8%)	4 (44.4%)	0.648
Restrictive VSD *n* (%)	2 (12.5%)	8 (88.9%)	0.021
Restrictive foramen ovale *n* (%)	0 (0.0%)	3 (33.3%)	0.043
AV/VA Discordance *n* (%)	6 (37.5%)	2 (22.2%)	1.000
Pulmonary Stenosis *n* (%)	5 (31.3%)	8 (88.9%)	0.014
Aortic Coarctation/ Hypoplasia *n* (%)	2 (12.5%)	1 (11.1%)	1.000
Reverse DV flow, *n* (%)	3 (18.8%)	5 (55.6%)	0.107
Ductus Venosus PI (Mean ± SD)	0.998 ± 0.25	1.34 ± 0.316	0.034
Umbilical Artery PI (Mean ± SD)	1.055 ± 0.269	0.936 ± 0.110	0.235
Middle Cerebral Artery PI (Mean ± SD)	1.95 ± 0.871	1.695 ± 0.605	0.478

TA: Tricuspid atresia, VSD: Ventricular septal defect, AV: Atrioventricular, VA: Ventriculoarterial, DV: Ductus venosus, PI: Pulsatility index.

**Table 4 diagnostics-14-02855-t004:** Gestational age and birth weight characteristics in survivors vs. non-survivors (liveborn pregnancies).

	Survivors (*n* = 16)	Non-Survivors (*n* = 6)	*p* Value
Gestational age at delivery (weeks), median (range)	37 (28–40)	36 (30–38)	0.412
Birth weight (g), median (range)	2550 (1200–3360)	2400 (1110–3200)	0.578

**Table 5 diagnostics-14-02855-t005:** Perinatal and postnatal outcomes by VA alignment.

Ventriculoarterial Connection	*n*	TOP (*n*)	Neonatal Death (*n*)	Survival at 3 Months	Survival at 12 Months
**Concordant**	16	3	3	10	10
None	6	0	-	6	6
Pulmonary stenosis	10	3	3	4	4
**Discordant**	8	-	2	6	6
None	2	-	0	0	2
Pulmonary stenosis	3	-	0	3	2
Aortic Coarctation/Hypoplasia	3	-	1	2	2
**Double outlet**	1	-	1	0	0
DOLV	1	0	1	0	0

VA: Ventriculoarterial, TOP: Termination of pregnancy, DOLV: Double-outlet left ventricle.

## Data Availability

The data presented in this study are available on request from the corresponding author.

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
