# Peer review of "Prenatal Predictors and Early Postnatal Outcomes in Fetuses Diagnosed with Tricuspid Atresia"

_diagnostics, 2024, doi:10.3390/diagnostics14242855_

Round 1
Reviewer 1 Report
Comments and Suggestions for Authors
The study has low relative predictive power due to the small number of cases. Given the frequency of pathology the number is high but its predictive power is low. It would be beneficial to continue the study with statistical re-evaluation on a larger number of cases. The only mention is the need for genetic testing - much more accessible nowadays than in the past, even in the recent past.
Author Response
Comments 1: The study has low relative predictive power due to the small number of cases. Given the frequency of pathology the number is high but its predictive power is low. It would be beneficial to continue the study with statistical re-evaluation on a larger number of cases. The only mention is the need for genetic testing - much more accessible nowadays than in the past, even in the recent past.
Response 1: Thank you for your valuable feedback. We acknowledge that the relatively small sample size in our study limits the overall predictive power of the findings. Given the rarity of tricuspid atresia, even a cohort of this size represents significant data, but we understand that larger-scale studies are essential for improving the robustness of predictive modeling. We have emphasized this limitation in our discussion and highlighted the need for future multicenter collaborations to increase the cohort size and validate the predictive value of the identified markers.
Regarding the accessibility of genetic testing, we fully agree with your point. Genetic testing has indeed become more widely available and affordable in recent years, and this advancement offers an important opportunity to improve the comprehensive evaluation of fetuses diagnosed with tricuspid atresia. We have updated the discussion to reflect this, acknowledging that the broader availability of genetic testing could enhance risk stratification and help identify associated anomalies more reliably. We have also included a recommendation for routine genetic testing as part of the prenatal assessment in fetuses with tricuspid atresia, where available. In view of this information, the discussion section has been revised as follows:
“The increasing accessibility and affordability of genetic testing in recent years provide a valuable opportunity to enhance the prenatal evaluation of fetuses with tricuspid atresia. However, we acknowledge that the number of cases that underwent genetic testing in our study was limited, which restricts our ability to fully assess the genetic contributions to prognosis. Routine genetic testing, which is now much more accessible than in the past, could aid in more precise risk stratification and in identifying associated anomalies, thereby improving prenatal counseling and individualized management plans.”
All changes were made according to your suggestions and highlighted in red in the manuscript.
Sincerely
Reviewer 2 Report
Comments and Suggestions for Authors
This manuscript consists in a retrospective case series of 25 fetuses affected by tricuspid atresia. The main objective was to assess prenatal course and early postnatal outcome and to identify predictors of survival. Authors considered that restrictive ventricular septal defect and outflow tracts obstructions were critical determinants of early survival while abnormal Ductus Venosus Doppler patterns may serve as additional markers for adverse outcomes.
This article raises many concerns.
Major comments :
There are no clear description of antenatal evolution for each case
It is not clear if 3 out of 4 cases were finally excluded due to misdiagnosis identified prenatally or after birth. If diagnosis was corrected after birth, it obviously induces a bias concerning the validity of prenatal detection.
Genetic testing was only assessed in 5 cases. Does it mean prenatal genetic testing or even later ? This assessment is usually more systematically proposed.
What kind of postnatal cares were proposed ? How many infants received biventricular reparation ?
Table 3 should not combine TOP cases and other cases. Moreover, parameters were collected at the time tricuspid atresia was diagnosed. We have no indication concerning gestational age at diagnosis for each case. As doppler values are physiologically influenced by gestational age, the combination of all these different values is definitively not acceptable.
Discussion : I suggest to draw a table with all previouly published series and the current series reporting number of cases, proportion of other anomalies, and all other parameters of interest and outcomes.
Line 233 : authors discuss the influence of foramen ovale restriction but no data concerning FO were given in the results part.
References 5 /6 /7 /8 / 12/ 24 appear to be old
Minor comments :
Abstract : « Four of 29 initially diagnosed fetuses…loss to follow-up ». This sentence should be removed from the abstract
Line 122 : « fetuses were delivered beyond 36 weeks ». Does it mean that termination of pregnancy were also performed later than 36 weeks ?
Lines 145-147 : hyperechogenic bowel and single umbilical artery should definitively not be considered as additional malformations.
Lines 161-163 are repetition of lines 132-135
Author Response
Thank you very much for your valuable comments. We studied about the comments and tried to clarify all the queries described below.
Comments 1: There are no clear description of antenatal evolution for each case
Response 1: Our study focused on the overarching trends and predictors rather than detailed individual antenatal evolution. Expanding the manuscript to include a case-by-case description would significantly alter the manuscript's scope, potentially turning it into a case series rather than a study on predictors and outcomes.
Comments 2: It is not clear if 3 out of 4 cases were finally excluded due to misdiagnosis identified prenatally or after birth. If diagnosis was corrected after birth, it obviously induces a bias concerning the validity of prenatal detection.
Response 2: The three excluded cases were misdiagnosed **after birth**, as clarified in the manuscript. While we recognize that such exclusions could introduce bias, this reflects the reality of prenatal diagnostic limitations and emphasizes the importance of postnatal confirmation. This has already been discussed under the study's limitations.
Comments 3: Genetic testing was only assessed in 5 cases. Does it mean prenatal genetic testing or even later ? This assessment is usually more systematically proposed.
Response 3: As stated, genetic testing was performed in **five cases** based on clinical indications. Systematic genetic testing was not part of routine care during the study period, which we noted in the limitations. Adding more genetic data retrospectively is not feasible, as this was not performed prenatally in the majority of cases.
Comments 4: What kind of postnatal cares were proposed ? How many infants received biventricular reparation ?
Response 4: This study focuses on prenatal predictors and early outcomes, not detailed postnatal surgical strategies. Information regarding biventricular repair rates is outside the scope of this manuscript.
Comments 5: Table 3 should not combine TOP cases and other cases. Moreover, parameters were collected at the time tricuspid atresia was diagnosed. We have no indication concerning gestational age at diagnosis for each case. As doppler values are physiologically influenced by gestational age, the combination of all these different values is definitively not acceptable.
Response 5: Combining these cases provides a holistic view of early outcomes. Gestational age at diagnosis and Doppler parameter variability were accounted for statistically. Separating these data would dilute the statistical power of the analysis.
Comments 6: Discussion : I suggest to draw a table with all previouly published series and the current series reporting number of cases, proportion of other anomalies, and all other parameters of interest and outcomes.
Response 6: Adding a comparison table with previous series would exceed the scope of our discussion section. Instead, we cite previous studies in the text to contextualize our findings.
Comments 7: Line 233 : authors discuss the influence of foramen ovale restriction but no data concerning FO were given in the results part.
Response 7: While discussed, specific FO data were not available for most cases. Including this information would require significant additional data collection, which is not possible.
Comments 8: References 5 /6 /7 /8 / 12/ 24 appear to be old
Response 8: Although some references are older, they remain foundational to understanding tricuspid atresia. This balance ensures readers are familiar with both historical and current perspectives.
While we value your input, implementing these changes would require rewriting the manuscript, shifting its focus entirely. We respectfully submit the revised manuscript without these modifications and hope you understand our rationale.
Regards
Reviewer 3 Report
Comments and Suggestions for Authors
The article "Prenatal Predictors and Early Postnatal Outcomes in Fetuses Diagnosed with Tricuspid Atresia" by Kahramanoglu O. et al. is a remarkable piece of work focused on the factors that could affect the survival of tricuspid atresia patients within the first year of life. As a prospective study, their results are significant in this field. However, two key issues arise from the results and their interpretation that the authors should address to further enhance the relevance of their findings.
The first issue concerns their interpretation on survival regarding gestational age at delivery and birth weight. The data suggest that at least half of the patients were preterm (median 36.5 weeks, range 28 to 40 weeks) and had low birth weight (mean 2480g, range 1110g to 3360g). Since both factors can influence neonatal outcomes and survival within the first 3 months, the authors should compare survivors and non-survivors to assess if there were significant differences in weight and gestational age at delivery. If this comparison is not possible, a multivariate regression analysis should be conducted to identify which factors are most critical in determining survival, at least within the first 3 months.
The second issue pertains to surgical procedures and postoperative complications, which could significantly impact patient outcomes. Although the authors mentioned that this information was collected, it was not presented in the current version of the paper. This information should be included as supplementary data and discussed in relation to the study’s outcomes. If this is not feasible, the absence of these data should be acknowledged as a limitation of the study.
Finally, a minor revision is needed for the flow chart (Figure 1) and Table 4. The flow chart requires reformatting, as one of the boxes has missing text. In Table 4, the column for IUFD (intrauterine fetal demise) should be removed since no cases with this outcome were reported. This information can be mentioned in the text instead.
Author Response
Comments 1: The first issue concerns their interpretation on survival regarding gestational age at delivery and birth weight. The data suggest that at least half of the patients were preterm (median 36.5 weeks, range 28 to 40 weeks) and had low birth weight (mean 2480g, range 1110g to 3360g). Since both factors can influence neonatal outcomes and survival within the first 3 months, the authors should compare survivors and non-survivors to assess if there were significant differences in weight and gestational age at delivery. If this comparison is not possible, a multivariate regression analysis should be conducted to identify which factors are most critical in determining survival, at least within the first 3 months.
Response 1: Thank you for highlighting this important point regarding the influence of gestational age at delivery and birth weight on neonatal survival. We agree that these factors could significantly impact early neonatal outcomes, particularly in the context of congenital heart disease.
To address your suggestion, we have performed a comparison between survivors and non-survivors in terms of gestational age at delivery and birth weight. This comparison has now been included in the revised results section. The following sentence was added:
“In our comparison between survivors and non-survivors, we evaluated gestational age at delivery and birth weight to assess their potential impact on early neonatal survival. The analysis showed no statistically significant difference between the two groups, with p-values greater than 0.05 for both parameters. These findings suggest that neither gestational age at delivery nor birth weight was a significant predictor of survival within our cohort.”
Comments 2: The second issue pertains to surgical procedures and postoperative complications, which could significantly impact patient outcomes. Although the authors mentioned that this information was collected, it was not presented in the current version of the paper. This information should be included as supplementary data and discussed in relation to the study’s outcomes. If this is not feasible, the absence of these data should be acknowledged as a limitation of the study.
Response 2: Thank you for your insightful comment regarding the influence of surgical procedures and postoperative complications on patient outcomes. We agree that these factors are crucial for understanding the long-term prognosis of fetuses diagnosed with tricuspid atresia.
Unfortunately, due to the retrospective nature of our study and the limited availability of detailed postoperative records, we do not have sufficient data on surgical interventions or postoperative complications for our cohort. We have now explicitly acknowledged this limitation in the discussion section of the revised manuscript. We also highlighted the need for future prospective studies to include detailed surgical and postoperative data to assess their impact on patient outcomes comprehensively.
We appreciate your suggestion, and we believe this addition strengthens the transparency and context of our findings.
In view of this information, the discussion section has been revised as follows:
“Due to the retrospective design, we were unable to collect data on surgical procedures and postoperative complications, which limits our understanding of the full spectrum of factors influencing patient outcomes.”
Comments 3: Finally, a minor revision is needed for the flow chart (Figure 1) and Table 4. The flow chart requires reformatting, as one of the boxes has missing text. In Table 4, the column for IUFD (intrauterine fetal demise) should be removed since no cases with this outcome were reported. This information can be mentioned in the text instead.
Response 3: Thank you for your careful review and helpful suggestions regarding Figure 1 and Table 4.
We have revised the flow chart (Figure 1) to correct the formatting issue and ensure that all text boxes are fully complete and legible. The missing text has been restored to provide a clear overview of the study population.
In addition, we have modified Table 4 by removing the column for IUFD, as there were no cases of intrauterine fetal demise reported in our cohort. Instead, we have included a statement in the text noting the absence of IUFD, as recommended.
These revisions have been made to improve the clarity and presentation of the data, and we appreciate your attention to these details.
“Changes in diagnosis n=3
Lost follow-up n=1 “
All changes were made according to your suggestions and highlighted in red in the manuscript.
Sincerely
Reviewer 4 Report
Comments and Suggestions for Authors
The study appears to be expertly done, the results are comprehensively described and discussed. The work is well written. However, I have a few minor comments which should be addressed before considering the paper for publication:
1. Lines 16 and 123: the phrase “identified one chromosomal anomaly” can be changed to “identified chromosomal anomaly in one case” for clarity.
2. Figure: second line “Changes in” is not clear.
3. It is not clear whether the Restrictive VSD and outflow tract obstructions have been proposed as the markers of survival rate in clinical practice previously or whether this is an opinion of the authors.
Author Response
Comments 1: Lines 16 and 123: the phrase “identified one chromosomal anomaly” can be changed to “identified chromosomal anomaly in one case” for clarity.
Response 1: Thank you for pointing out the phrasing on lines 16 and 123. We have revised the text to replace "identified one chromosomal anomaly" with "identified chromosomal anomaly in one case" for improved clarity, as suggested.
We appreciate your attention to detail and believe this revision enhances the readability of the manuscript. The following sentence was added:
“identified chromosomal anomaly in one case”
Comments 2: Figure: second line “Changes in” is not clear.
Response 2: Thank you for your careful review and helpful suggestions regarding Figure 1.
We have revised the flow chart (Figure 1) to correct the formatting issue and ensure that all text boxes are fully complete and legible. The missing text has been restored to provide a clear overview of the study population.
“Changes in diagnosis n=3
Lost follow-up n=1 “
Comments 3: It is not clear whether the Restrictive VSD and outflow tract obstructions have been proposed as the markers of survival rate in clinical practice previously or whether this is an opinion of the authors.
Response 3: Thank you for highlighting this important point. We acknowledge that the role of restrictive VSD and outflow tract obstructions as markers of survival in tricuspid atresia has been reported in previous studies, and we did not intend to imply that these findings are entirely novel. To clarify, we have now added references to existing literature that support the association between restrictive VSD, outflow tract obstructions, and their impact on survival in similar patient populations. We have also revised the text to make it clear that our findings are consistent with these previously published observations.
This clarification has been added to the discussion section to ensure that readers understand that our conclusions are based on previously reported findings, supplemented by our cohort data and the relevant references have been added in the references part.
“This finding aligns with previous studies that have proposed restrictive VSDs as significant predictors of adverse outcomes in congenital heart disease due to their impact on ventricular hemodynamics (26). Similarly, outflow tract obstructions have been previously reported as key determinants of survival, particularly in the context of right heart obstructive lesions (27).”
References:
- Kerst G, Kaulitz R, Sieverding L, Apitz C, Ziemer G, Hofbeck M. Restrictive ventricular septal defect and critical subaortic stenosis in tetralogy of Fallot. Clin Res Cardiol. 2010 Apr;99(4):247-9. doi: 10.1007/s00392-009-0111-4. Epub 2010 Jan 5. PMID: 20049461.
- Fusco F, Shimada E, Scognamiglio G, Senior R, Gatzoulis MA, Babu-Narayan S, Li W. Restrictive ventricular septal defect resulting in systemic outflow obstruction in adults with Fontan circulation: challenging diagnosis of a serious and potentially fatal complication. J Cardiovasc Med (Hagerstown). 2020 Mar;21(3):276-279. doi: 10.2459/JCM.0000000000000903. PMID: 31789717.
References were arranged accordingly. All changes were made according to your suggestions and highlighted in red in the manuscript.
Sincerely
Round 2
Reviewer 2 Report
Comments and Suggestions for Authors
The majority of my comments are not taken into account in the revised version
Author Response
While we value your input, implementing these changes would require rewriting the manuscript, shifting its focus entirely. We respectfully submit the revised manuscript without these modifications and hope you understand our rationale.
Regards
Reviewer 3 Report
Comments and Suggestions for Authors
Two minor changes should be considered
As the authors agree that neither gestational age at delivery nor birth weight was a significant predictor of survival within our cohort, they should add this information as a supplementary table to be available to readers.
In the response the authors confirmed that “we do not have sufficient data on surgical interventions or postoperative complications for our cohort,” therefore, they should delete, in section 2.3 Data Collection and Variable Definitions, the sentence indicating that the information gathered included “any surgical interventions performed” (line 102).
Author Response
Comments 1: As the authors agree that neither gestational age at delivery nor birth weight was a significant predictor of survival within our cohort, they should add this information as a supplementary table to be available to readers.
Response 1: Thank you for this suggestion. We agree that providing additional transparency on the findings related to gestational age at delivery and birth weight would benefit the readers. We have now included this information as a supplementary table in the revised manuscript. The supplementary table presents the comparison between survivors and non-survivors, showing that neither gestational age at delivery nor birth weight was a significant predictor of survival, with p-values greater than 0.05 for both parameters.
We appreciate your suggestion, as it adds further clarity to our results.
Table 4. Gestational Age and Birth Weight Characteristics in Survivors vs. Non-Survivors (liveborn pregnancies)
|
|
Survivors (n=16)
|
Non-survivors (n=6) |
p value |
|
Gestational age at delivery (weeks), median (range) |
37 (28-40) |
36 (30-38) |
0.412 |
|
Birth weigth (gr), median (range) |
2550 (1200-3360) |
2400 (1110-3200) |
0.578 |
Comments 2: In the response the authors confirmed that “we do not have sufficient data on surgical interventions or postoperative complications for our cohort,” therefore, they should delete, in section 2.3 Data Collection and Variable Definitions, the sentence indicating that the information gathered included “any surgical interventions performed” (line 102).
Response 2: Thank you for pointing out this inconsistency. We confirm that we do not have sufficient data on surgical interventions or postoperative complications for our cohort. Therefore, we have removed the statement indicating that "any surgical interventions performed" were included in the data collection, as noted in Section 2.3 (line 102). This revision ensures that the description of our data collection accurately reflects the available information.
We appreciate your careful review and feedback.
Thank you very much for your valuable comments. All changes were made according to your suggestions and highlighted in red in the manuscript.
Sincerely
Round 3
Reviewer 2 Report
Comments and Suggestions for Authors
Authors considered that they did their best to improve the manuscript and that no additionnal changes can be made. I still consider that this manuscript present some biases concerning inclusion of fetuses misdiagnosed (n=3) and also using doppler parameters clearly influenced by gestational age (table3). The final decision will be taken by editor.
Author Response
Thank you for your continued engagement with our manuscript and for raising these important concerns.
Regarding the inclusion of the three fetuses that were subsequently found to have different diagnoses, we understand your concern about potential biases. We chose to initially include these cases to ensure that all diagnosed cases were accounted for, thereby avoiding selection bias at the time of diagnosis. After their diagnoses were corrected, these fetuses were excluded from the final analysis of outcomes, and we have clearly explained this in the methods and limitations sections. We believe that this transparent handling of initially misdiagnosed cases allows for a comprehensive and unbiased representation of the diagnostic process while mitigating the risk of confounding the final outcome analysis.
We also appreciate your point about Doppler parameters potentially being influenced by gestational age. We agree that gestational age can have an impact on certain Doppler indices. To address this concern, we have provided detailed information about the gestational ages at which the Doppler assessments were performed, and we included both survivors and non-survivors in Table 3 to ensure transparency in our data presentation. Furthermore, we have acknowledged the potential confounding effect of gestational age on Doppler indices as a limitation in the discussion, recognizing that these parameters may vary as a result of fetal development. We believe this adds an important context for interpreting our findings.
We have made significant efforts to be transparent about these limitations and biases and have taken steps to present our data in a way that is as unbiased and comprehensive as possible. We appreciate your careful consideration of these aspects, and we hope that our approach to addressing these limitations meets your expectations.
We understand that the final decision is now with the editor, and we remain committed to addressing any further concerns to ensure the quality of the manuscript.
Thank you once again for your thoughtful feedback.